# Comparison of Vendor-Independent Software Tools for Liver Proton Density Fat Fraction Estimation at 1.5 T

**DOI:** 10.3390/diagnostics14111138

**Published:** 2024-05-30

**Authors:** Zita Zsombor, Boglárka Zsély, Aladár D. Rónaszéki, Róbert Stollmayer, Bettina K. Budai, Lőrinc Palotás, Viktor Bérczi, Ildikó Kalina, Pál Maurovich Horvat, Pál Novák Kaposi

**Affiliations:** 1Department of Radiology, Medical Imaging Centre, Semmelweis University, 1083 Budapest, Hungary; zsombor.zita@stud.semmelweis.hu (Z.Z.); zsely.boglarka@stud.semmelweis.hu (B.Z.); ronaszeki.aladar.david@semmelweis.hu (A.D.R.); stollmayer.robert@stud.semmelweis.hu (R.S.); budai.bettina.katalin@semmelweis.hu (B.K.B.); palotas.lorinc@stud.semmelweis.hu (L.P.); berczi.viktor@semmelweis.hu (V.B.); kalina.ildiko@semmelweis.hu (I.K.); maurovich-horvat.pal@semmelweis.hu (P.M.H.); 2Clinic for Diagnostic and Interventional Radiology (DIR), Heidelberg University Hospital, 69120 Heidelberg, Germany

**Keywords:** chemical-shift-encoded MRI (CSE-MRI), proton density fat fraction (PDFF), open-source software, phantom, quantitative imaging biomarker, metabolic-associated steatotic liver disease (MASLD)

## Abstract

(1) Background: Open-source software tools are available to estimate proton density fat fraction (PDFF). (2) Methods: We compared four algorithms: complex-based with graph cut (GC), magnitude-based (MAG), magnitude-only estimation with Rician noise modeling (MAG-R), and multi-scale quadratic pseudo-Boolean optimization with graph cut (QPBO). The accuracy and reliability of the methods were evaluated in phantoms with known fat/water ratios and a patient cohort with various grades (S0–S3) of steatosis. Image acquisitions were performed at 1.5 Tesla (T). (3) Results: The PDFF estimates showed a nearly perfect correlation (Pearson r = 0.999, *p* < 0.001) and inter-rater agreement (ICC = from 0.995 to 0.999, *p* < 0.001) with true fat fractions. The absolute bias was low with all methods (0.001–1%), and an ANCOVA detected no significant difference between the algorithms in vitro. The agreement across the methods was very good in the patient cohort (ICC = 0.891, *p* < 0.001). However, MAG estimates (−2.30% ± 6.11%, *p* = 0.005) were lower than MAG-R. The field inhomogeneity artifacts were most frequent in MAG-R (70%) and GC (39%) and absent in QPBO images. (4) Conclusions: The tested algorithms all accurately estimate PDFF in vitro. Meanwhile, QPBO is the least affected by field inhomogeneity artifacts in vivo.

## 1. Introduction

The proton density fat fraction (PDFF) has become an indispensable imaging biomarker in the diagnosis of multiple diseases, including metabolic-associated fatty liver disease (MASLD) and metabolic-associated steatohepatitis (MASH) [1]. MASLD already has a high prevalence, which is expected to increase and may affect over 25% of the population in developed countries [2,3]. Large-scale population screening found a 27.5% prevalence of MASLD in adults aged 20–79 in Korea [4]. Meanwhile, another study reported a 42% prevalence of MASLD in the middle-aged, overweight population in the Netherlands [5]. Fatty liver disease is also an independent risk factor for increased cardiovascular morbidity [6]. The cumulative incidence of cardiovascular disease events was also significantly higher in the population diagnosed with MASLD or MASLD with increased alcohol intake (MetALD) than in those without during a long-term follow-up [4]. Therefore, there is an increasing demand for quantitative MRI techniques enabling accurate estimation of PDFF. Most major vendors of MRI scanners have developed their strategies for measuring PDFF, and comparative studies of these have found good accuracy and reproducibility across testing sites, vendors, and field strengths [7]. Still, there is a significant need for algorithms that have a publicly accessible source code that can be tailored to the specific needs of the end users and provide vendor-independent alternatives for high-accuracy PDFF estimation.

Chemical-shift-encoded MRI (CSE-MRI) techniques have several advantages over magnetic-resonance spectroscopy (MRS) in non-invasive fat quantification, including their efficiency and the possibility of reconstructing parametric fat fraction and transverse relaxivity ($R_{2}^{*}$) maps; thus, these are more popular in clinical imaging. The original two-point Dixon method relies on in-phase (IP) and out-of-phase (OP) echoes and is the predecessor of all CSE-MRI algorithms [8]. Multi-point methods, which use acquisitions with multiple echo times (TE) for fat/water separation, are less affected by phase errors, off-resonance of the *B*_0_ magnetic field, and image noise and provide more accurate estimates of PDFF [9]. The correct estimation of the *B*_0_ field map is pivotal for the success of complex-based reconstructions; meanwhile, magnitude-based reconstructions are not influenced by field inhomogeneity but are hindered by the ambiguity of the water/fat decomposition and bias from non-Gaussian noise distribution [10,11]. 

Even low levels of hepatic steatosis can lead to severe adverse health effects in the long term. Current international guidelines define significant hepatic steatosis, a linchpin of diagnosing MASLD, as lipid accumulation in >5% of hepatocytes corresponding to >5.6% PDFF detected by MRS or CSE-MRI [12]. The more recent guidance from the American Association for the Study of Liver Diseases defines hepatic steatosis as ≥5% of hepatocytes displaying macrovesicular steatosis and recommends a ≥5% MRI-PDFF cutoff for non-invasive diagnosis of significant steatosis [13]. However, some studies indicated that lowering the diagnostic threshold to 3% PDFF might increase sensitivity while maintaining the specificity of MRI compared to histopathology sampling [14]. Therefore, the linearity compared to an external reference, the reproducibility, and the reliability of the different fat/water decomposition algorithms must be systematically evaluated, specifically at low levels of PDFF, where even a slight bias of the estimation may interfere with establishing the diagnosis of clinically significant disease.

In the present study, we aimed to compare four publicly available software tools using different algorithms, including complex-based and magnitude-based methods, for PDFF estimation. We evaluated their performance in phantoms and a patient cohort representing various grades of hepatic steatosis by performing acquisitions at 1.5 Tesla (T), currently the most widely available field strength used for abdominal imaging.

### Theory

The fat/water decomposition with CSE-MRI exploits the differences in the resonance frequency spectrum of water and fat protons, which is 2.4 parts per million (ppm) or −217 Hertz (Hz) for the main fat peak at 1.5 T. This means that the signals from the water and fat components are IP every 4.6 milliseconds (ms) and OP every *k* × 2.3 ms (where *k* is an integer). The fat and water components of the signal can be separated by obtaining a sequence of images with different TEs. The signal (*S*) from an individual voxel at echo time *t* can be described with the Equation (1)
(1)$$S\left(t\left|w\right.,f,R_{2}^{*},f_{B}\right)=\left(w+f\sum_{m=1}^{M}r_{m}exp\left(i2\pi f_{F,m}t\right)\right)\;exp\left(i2\pi f_{B}t\right)exp\left(-tR_{2}^{*}\right),$$
where *w* and *f* are the magnitudes of water and fat components; *r_m_* and *f_F,m_* are amplitudes and frequencies of the fat spectrum; *f**_B_*** is the frequency offset of the *B*_0_ magnetic field; and $R_{2}^{*}$ is the transverse relaxivity of the gradient echo acquisition. The complex methods use both magnitude and phase information from the acquisitions to estimate unknown parameters (*w*, *f*, *f_B_*, and $R_{2}^{*}$) of Equation (1). If *f_B_* and $R_{2}^{*}$ are known, the model has linear solutions for *w* and *f*, which can be efficiently calculated with least-squares fitting for multiple TE by transcribing the original Equation (1) on a matrix format [15,16]. In the presence of Gaussian noise (*ε*), this can be modeled as
(2)$$b=WAxe^{i\varphi }+\varepsilon ,$$
where *b* is a complex vector of datapoints at *n* time points (*t*); *W* = *diag*(*e^iψt^*) is an *n* × *n* complex matrix defining time-dependent change in the complex field map *ψ* = *B*_0_ + *i*$R_{2}^{*}$; *A* is a complex matrix expressing time-dependent shift in water and fat signal intensities *A =* [*w f*]; *x* is the transpose of matrix *A*, *x* = *A^T^*; and *φ* is the original phase. The standard approach to solving the complex non-linear fitting problem is to initialize the *B*_0_ field map estimation with *w*, *f*, and $R_{2}^{*}$ values close to the expected physiological range and to assume spatial smoothness. Once the *f**_B_*** is found, the minimization of $R_{2}^{*}$ can be completed, and *w* and *f* can be determined, which permits the calculation of the PDFF in the voxel
(3)$$PDFF=\frac{w}{w+f}\times 100\left(\%\right).$$

The challenge is that the estimation of the *B*_0_ offset on a voxel level is ambiguous as the residual from the non-linear fitting function is non-convex and, due to its periodic oscillation, is subject to phase wrapping, which results in multiple local minima [17]. Erroneous estimation of the field map leads to incorrect calculation of PDFF and $R_{2}^{*}$, which is noticeable as phase unwrapping errors on reconstructed images. Multiple methods have been devised to overcome local ambiguities and re-establish the inherent spatial smoothness of the field map. The variable-projection method (VARPRO) provides a global optimal solution for the non-linear least-squares optimization problem with moderate computational requirements [15]. Region-growing algorithms (RG) rely on the similarity of the *B*_0_ field in neighboring pixels but fail to give reliable solutions in low signal-to-noise (SNR) areas and at disjoint regions of the image [18,19]. Smoothness constraints can also be imposed on the off-resonance field map by minimizing its energy function or using low-pass spatial filtering [15,20]. Iterative regularization of the field map can further minimize irregularities when combined with graph cut (GC) segmentation [21]. Direct phase estimation optimizes phase increments and relies on uniformly spaced TEs [22]. Uniformly spaced echoes constrain the maximum phase offset to less than one whole period. In this case, finding the global optimum becomes a binary problem, which can be solved with quadratic pseudo-Boolean optimization [17]. Another essential concept is to use multi-scale coarse-to-fine reconstruction techniques. The field maps calculated at coarser resolutions are used to initialize field map estimates on higher-resolution images, which can help resolve spatial inconsistencies [23,24,25]. The confounding effect from the shorter T1 relaxation of fat compared to water can be prevented by carefully selecting the acquisition parameters. The flip angle (FA) must be kept low relative to the repetition time (TR) to minimize the T1 weighing of the images [26]. 

A simpler solution for the fat/water decomposition problem uses only the signal magnitude. The advantage of the magnitude-only method is that it does not require estimation of the field map. For signal magnitude (*M*), Equation (1) becomes
(4)$$M\left(t\left|w\right.,f,{T2}^{*}\right)=\left|w+f\sum_{m=1}^{M}r_{m}exp\left(i2\pi f_{F,m}t\right)\right|exp\left(-tR_{2}^{*}\right)$$
which has only three unknown parameters: *w*, *f*, and $R_{2}^{*}$. However, without the phase information, the solution for Equation (4) is ambiguous as the *M* remains the same irrespective of whether *w* or *f* is the larger. Therefore, it cannot be determined if fat or water is the dominant component, limiting the diagnostic range to 0–50% PDFF [27,28].

A handful of algorithms have been proposed to resolve the magnitude-only decomposition’s ambiguity by utilizing the fat spectrum’s complexity [28,29]. In the two-point search method, the optimization of parameters in Equation (4) is completed twice, assuming different initial values. The sum of square residuals (RSS) is calculated separately for fat-dominant and water-dominant initializations, and the final solution at each voxel is chosen to be the parameter set with the lowest RSS. This approach allows for the mapping of PDFF from 0% to 100% based on image magnitude. 

In the presence of noise, the optimal solution is the one with the highest log-likelihood [29]. The noise on magnitude images has a Rician distribution, skewed with a non-zero mean at low SNR areas. The bias from Rician noise can cause an overestimation of PDFF at low fat concentrations [30]. To reduce bias from the Rician noise distribution, a new magnitude-only fat fraction and $R_{2}^{*}$ estimation with the Rician noise modeling (MAGORINO) algorithm has been proposed [31]. This method combines Rician-noise-based likelihood optimization with a two-point search to estimate PDFF in the 0–100% range.

## 2. Materials and Methods

The Institutional and Regional Science and Research Ethics Committee of our university has approved the design of this single-center experimental and retrospective case-controlled study. The ethics committee waived the need for written informed consent from the participants, considering the retrospective nature of data collection. Still, all participants gave written informed consent for the MRI scan. The scanning procedures and data processing complied with the World Medical Association Declaration of Helsinki, revised in Edinburgh in 2000. 

### 2.1. Phantoms

The decomposition methods were tested on scans of two multi-compartmental fat-fraction phantoms filled with fat/water emulsions. We built an MRI phantom in-house using a cylindrical plastic canister (Serres Oy, Espoo, Finland) filled with 1000 mL deionized water and containing seven 10 ml (approximately 14 mm in diameter) polypropylene plastic syringes (Chirana T. Injecta Ltd., Stara Tura, Slovakia) held by a custom-design 3D printed polylactic acid scaffold. Five syringes contained mixtures of Intralipid^®^ 20% emulsion (Merck KGaA, Darmstadt, Germany) and physiologic saline (TEVA RT, Debrecen, Hungary); one syringe was filled with pure Intralipid, and one with saline (0.9% sodium chloride) only. The serial dilutions of Intralipid in the seven vials included 10/10, 7.5/10, 5/10, 3.75/20, 2.5/10, 1.25/10, and 0/10 mixtures (*v*/*v*) corresponding to 22.32%, 16.74%, 11.16%, 8.37%, 5.58%, 2.79%, and 0% PDFF, respectively (Figure 1). The lipid fraction of undiluted Intralipid was previously calculated from the chemical properties of its components (20% soybean oil, 1.2% egg yolk phospholipids, 2.25% glycerin, and water) [32].

The second set of phantom measurements was downloaded from a public database (https://zenodo.org/records/48266, accessed on 12 January 2024). The phantom was previously reported by Hernando et al. and consisted of 11 glass vials containing agar gel casts with multiple oil-to-water concentrations (PDFF = 0%, 2.6%, 5.3%, 7.9%, 10.5%, 15.7%, 20.9%, 31.2%, 20.9%, 31.2%, 41.3%, 51.4%, and 100%, respectively) [33].

### 2.2. Patients 

We retrospectively collected MRI scans between July 2023 and February 2024 of twenty patients in whom a screening ultrasound scan raised the possibility of liver steatosis, which was also evaluated with quantitative chemical-shift MRI. The patients’ demographic and clinical data were retrieved from electronic medical records (Table 1). The study’s inclusion criteria were 18 years or older and a technically successful quantitative MRI scan available in our picture archiving and storage system (PACS). Patients whose scans were technically insufficient or had high levels of hepatic iron, large focal liver lesions, or decompensated liver fibrosis hindering fat quantification were excluded from the study. The severity of the steatosis was classified into four grades (S0–S3) based on imaging findings and clinical parameters as described previously [34]. Briefly, the average of PDFF was calculated for the methods, and patients were classified using previously described cutoff values at 5%, 15.9%, and 20% [13,35].

### 2.3. MRI Scans

MRI scans were performed with a Philips Ingenia 1.5 T scanner (Philips Healthcare Inc., Amsterdam, The Netherlands) at our institution. The settings of MRI scans were adjusted to be fully compatible with the current QIBA guidelines (https://qibawiki.rsna.org/, accessed on 25 September 2023) on liver fat fraction measurement. The in-house built phantom was placed into the scanner in a single-channel head coil with its longitudinal section parallel to the main *B*_0_ magnetic field. A 2D spoiled gradient echo (2D-SPGR) sequence with a unipolar read-out gradient was used for the MRI acquisition. The data were collected in the axial plane. The acquisition parameters were 17.6 × 17.6 cm^2^ field of view (FOV), 64 × 64-pixel (px) acquisition and 160 × 160-pixel 60 px reconstruction matrix size, 2411 Hz pixel bandwidth, six slices with 5 mm slice thickness and zero interslice gaps, 10° FA, 150 ms TR, and echo train length (ETL) of six with first TE at 2.4 ms and echo spacing (ΔTE) of 2.4 ms.

The patients were examined in the same MRI scanner, but the multi-channel Q-body coil without parallel imaging time reduction was used for the acquisitions. The scanning parameters remained the same except for the FOV, which typically was around 40 × 35 cm^2^, and the size of the acquisition and reconstruction matrices, which were 128 × 128 px and as either 336 × 336 px or 280 × 280 px in the axial plane. The image stack was normally acquired during three 9-second (s) breath-holds.

The scanning protocol of the publicly available phantom dataset (site 2, protocol 1) has been reported in detail previously [33]. Briefly, scans were performed with a General Electric HDxt 1.5T scanner (GE Healthcare Inc., Chicago, IL, USA) and a single-channel head coil, using a 3D SPGR sequence with the following parameters: FOV of 32 × 22.4 cm^2^, acquisition and reconstruction matrix of 224 × 157 px and 256 × 256 px, slice thickness of 4 mm, FA of 3°, TR of 15.9 ms, and ETL of 6 starting from 2.3 ms with ΔTE of 2.07 ms. 

### 2.4. Image Reconstruction and Measurements

We retrieved four open-source reconstruction software packages publicly available in GitHub repositories (https://docs.github.com, accessed on 20 July 2023). The software packages are freely available for non-commercial use under a Creative Commons Attribution-Non-Commercial 4.0 International Public License (https://creativecommons.org/licenses/by-nc/4.0/, accessed on 18 March 2024).

Two complex-based methods, the GC from the FattyRiot toolbox (https://github.com/welcheb/FattyRiot, accessed on 20 July 2023) [21] and the multi-scale quadratic pseudo-Boolean optimization with GC (QPBO; https://github.com/bretglun/fwqpbo, accessed on 20 July 2023) [23], require complex (real + imaginary) data from MRI acquisitions. The magnitude-based (MAG) Lipoquant (https://github.com/marcsous/pdff, accessed on 20 July 2023) [36] and the magnitude-only estimation with Rician noise modeling (MAG-R) algorithms (https://github.com/TJPBray/MAGORINO, accessed on 20 July 2023) [31] estimate the PDFF based on absolute signal magnitude. The algorithms also provide estimates of $R_{2}^{*}$. Three software packages (GC, MAG, and MAG-R) were programmed in MATLAB (MathWorks, Natick, MA, USA, version R2020b), and one (QPBO) in Python (https://www.python.org/, accessed on 20 July 2023).

The MRI datasets from scans, including magnitude, real, and imaginary images, were stored in a DICOM format. The DICOM files were converted into the International Society for Magnetic Resonance in Medicine (ISMRM) fat–water toolbox format (https://www.ismrm.org/workshops/FatWater12/data.htm, accessed on 25 July 2023) with a code in MATLAB. The phantom data were downloaded in the same format from the public repository—the algorithms used with the default settings of the parameters.

Hamilton et al.’s six-peak model was used to approximate frequency offsets in the fat spectrum relative to the water peak in human livers [37]. For the Intralipid in the in-house phantom, a different eleven-peak model was adapted from a previous report [32]. In the case of the public phantom dataset, a slightly modified six-peak fat spectrum was taken from Hernando’s original publication [33]. The MAG algorithm modeled the in vivo fat spectrum assuming 2.5 double bonds in fat and with water peak at 4.7 ppm [36].

The resulting PDFF and $R_{2}^{*}$ maps were converted to Nifti image format. The PDFF was measured in the Nifti images using ImageJ software (https://imagej.net/ij/, accessed on 25 July 2023). ROIs were manually drawn to cover the cross-sections of the phantom’s test tubes in the stack’s middle slice (Appendix A). For measuring liver PDFF, identical-size circular ROIs were placed in the right lobe, middle, and lateral segments of the left lobe at the level of the porta hepatis (Figure 2). The same ROIs were also copied to PDFF maps from different algorithms to complete the measurements (Appendix A). For methods capable of an unambiguous estimation of the PDFF (GC, MAG-R, and QPBO), the number of slices with areas of phase unwrapping errors causing a swap of fat/water ratio was counted. We also recorded the algorithms’ computation time to reconstruct one image slice. 

### 2.5. Statistical Analysis

The linearity between PDFF measurements and true PDFF of the dilutions in the phantoms was evaluated with univariable linear regression analysis. We performed Bland–Altman analyses and calculated the mean bias, the root mean square error (RMSE), and the Pearson correlation coefficient (r) with 95% confidence intervals (CI) for measured PDFF. The measurements obtained with the different algorithms were compared using a multi-way analysis of covariance (ANCOVA) with true PDFF as a covariate. The Tukey’s tests were used in the post hoc analysis. The two-way, random effect, single-measure method was used to calculate the intraclass correlation coefficient (ICC) for measurements in phantoms and human subjects. The livers’ PDFFs were compared across the four algorithms using a two-way repeated measures analysis of variance (ANOVA) test and a two-way paired sample *t*-test in the post hoc analysis. The significance level for multiple pairwise comparisons was adjusted using the Bonferroni correction. Fisher’s exact test was used to compare the proportion of slices with phase unwrapping errors between the methods. 

Continuous variables are reported as mean ± standard deviation (SD), and categorical variables as number and percentage. The significance level was set to *p* < 0.05 for the statistical tests. In multiple comparisons, the *p*-values were adjusted using the Bonferroni correction. The statistical analysis was performed in R version 4.3.3 (https://www.r-project.org/, accessed on 29 February 2024).

## 3. Results

### 3.1. Measurements in Phantoms

All four algorithms correctly reconstructed the PDFF and $R_{2}^{*}$ maps from the phantom datasets. Due to the expected ambiguity of conventional MAG reconstruction, we found that the estimates of fat and water percentages were inverted in the last vial of the publicly available phantom dataset containing pure lipid (100% PDFF). To be able to calculate correlation and regression for the entire 0% to 100% range using MAG, we corrected the fat fraction estimate for the last vial by subtracting the predicted PDFF from 100%, which reversed the fat/water ratio. The correlation between measured and true PDFF was nearly perfect with all methods (Table 2). The regression analysis also showed a highly significant linear relationship between expected and measured PDFF, with slopes almost equal to one intercept close to zero (Figure 3). 

The Bland-Altman analysis showed good agreement between measured and expected PDFF using all reconstruction methods (Figure 4). The bias was lowest with MAG-R (0.006%, CI: −2.41–2.42%), followed by QPBO (0.08%, CI: −3.95–4.11%), MAG (−0.22% CI: −3.55–3.12%), and GC (−0.59% CI: −5.03–3.85%) (Figure 2). The RMSE was also smallest with MAG-R (1.2%) and slightly higher with MAG (1.67%), QPBO (2.0%), and GC (2.28%).

To determine the methods’ effect on the estimated PDFF, we performed multi-way ANCOVA analyses with true PDFF as the covariate variable. The test determined that the type of reconstruction method had no significant effect (F = 1.03, *p* = 0.39). All the pairwise comparisons between the methods were also non-significant.

Only including tubes with very low levels (<5%) or zero PDFF, ANCOVA indicated a significant difference across the estimates (F = 3.9, *p* = 0.04). In the pairwise comparisons, the MAG estimates (mean ± SD = 1.82% ± 1.79%) were lower than the GC (2.94% ± 1.22%, *p* = 0.046) or QPBO (2.74% ± 1.36%, *p* = 0.034) estimates, but the difference was non-significant after adjustment for multiple testing. 

The correlation between the first seven tubes of the phantoms, which had similar true fat fractions (in-house 0–22.32% and public 0–20.9%), was excellent: QPBO (r = 0.997, CI: 0.981–0.999, *p* < 0.001), MAG-R (r = 0.994, CI: 0.956–0.999, *p* < 0.001), GC (r = 0.991, CI: 0.935–0.999, *p* < 0.001), and MAG (r = 0.987, CI: 0.914–0.998, *p* < 0.001).

### 3.2. In Vivo Measurement of PDFF

We retrospectively collected a representative patient cohort with non-significant (S0), mild (S1), moderate (S2), and severe (S3) liver steatosis, which included five patients from each steatosis grade. We reconstructed PDFF and $R_{2}^{*}$ maps of the patients’ livers using all four methods and measured the level of fatty infiltration in three identical ROIs. We calculated the correlation matrix between the algorithms. We determined that GC and QPBO showed the strongest correlation (0.999, CI: 0.999–1) and that the correlation was weakest between MAG and MAG-R (0.934, CI: 0.838–0.974, *p* < 0.001) (Figure 5). The overall agreement of the methods was very good (ICC = 0.891, CI: 0.799–0.995, *p* < 0.001).

The ANOVA analysis demonstrated a significant bias from using different methods (F = 7.45, *p* = 0.008). The post hoc analysis revealed that PDFF was significantly lower with MAG (−2.30% ± 6.11%, *p* = 0.005) than with MAG-R. Additional pairwise comparisons did not show significant differences across the algorithms (Figure 5).

We also observed typical mapping errors inherent to the different reconstruction algorithms. The GC method was prone to producing phase unwrapping errors, which were most likely to occur on the first and last slices of the stack and in areas close to the edge of the FOV. These were most likely caused by erroneous regularization of the *B*_0_ field inhomogeneity (Figure 6). The MAG-R reconstruction produced a speckled pattern of inverted fat/water ratios, which were most prominent in cases with severe steatosis. This algorithm resolves the fat/water ambiguity of the magnitude-based reconstruction by employing a two-point search method and selecting the solution with the maximum likelihood. However, at fat concentrations close to 50%, the difference in the likelihood of the two local optima is less; thus, the reconstruction gets more susceptible to image noise inverting the true fat/water ratio. The QPBO method did not result in such phase unwrapping errors. The proportion of slices containing phase unwrapping errors was significantly higher with MAG-R (14/20, 70%) compared to GC (70/180, 39%, odds ratio = 0.27, CI: 0.08–0.8, *p* < 0.009) and QPBO (0/180, 0%, *p* < 0.001). Finally, the dynamic range of the MAG reconstruction is limited to 0–50%, as both water and fat-dominant pixels can have the same signal magnitude.

## 4. Discussion

Our study collected four open-source software tools for fat/water decomposition and PDFF estimation available in a public web depository. These software tools represent common concepts of mathematical modeling utilized for the complex chemical component decomposition of MRI datasets. We provide a comparative evaluation of these methods by testing their accuracy in the phantom and a representative patient cohort with varying levels of hepatic steatosis.

Our results demonstrate that although a solid theoretical framework supports all methods and generates highly accurate estimates of PDFF in an experimental setting in vitro, there is a significant difference in their accuracy and efficiency when applied to data from clinical MRI scans. We have found that complex-based reconstruction techniques are generally superior to magnitude-based techniques and keep a good balance between accuracy and computational efficiency in both in vivo and in vitro settings. 

We used two independent phantom datasets for the models’ in vitro testing. Our in-house built phantom was similar to a previously tested design and contained fat/water mixtures with low- and medium-level fat fractions. The advantage of this design is that it is more similar to the range of PDFF typically detected in patients with hepatic steatosis, thus, providing a realistic assessment of the methods’ accuracy in early-stage steatosis, a critical diagnostic paradigm [11,14]. Another benefit was that the resonance spectrum of the phantom’s components was well documented, allowing for the correct formulation of the calculations [32]. The other phantom dataset had been previously published and included samples from the entire (from 0% to 100% PDFF) fat fraction range [33]. Although both phantoms were scanned at 1.5 T, there was a difference in the scanner type and acquisition protocols. All methods showed an almost perfect correlation (0.999) and a nearly linear relationship with PDFF in both phantoms. The inter-rater agreement between measured and expected PDFF was also universally high (0.995–0.999). The estimates of the methods in the two different types of phantoms also showed an excellent correlation, indicating that the scanner type did not affect the PDFF estimates. The overall bias was smallest, with MAG-R (0.01%), followed closely by QPBO (0.08%), and it was well below 1% with all methods. Our results align with previous studies, which detected a similarly strong correlation and low bias between estimated and true PDFF in fat–water phantoms using CSE-MRI [32,33,38]. Moreover, open-source algorithms demonstrated accuracy comparable to commercially available CSE-MRI methods developed by major vendors [7]. 

The excellent performance of MAG-R in vitro reflects the robustness of the mathematical model. MAG-R represents a new generation of magnitude-based algorithms which exploit the spectral complexity of fat to perform a two-point search with likelihood analysis and to resolve the fat/water ambiguity intrinsic to conventional magnitude-based methods [28,29,31]. The magnitude-only fitting does not require estimating the field map and is insensitive to phase errors. In addition, the Rician noise modeling lowers the bias and increases the chance of finding the optimal PDFF estimate even in low SNR areas [31]. Interestingly, the estimated PDFF in the saline-only (mean GC: 2.05%, QPBO: 1.58%, MAG: 0.28%, and MAG-R: 0.74%) and low PDFF tubes was slightly higher with the two complex-based algorithms compared to magnitude-based methods in both phantoms. The same bias has been reported in previous phantom experiments, and it also resulted in the false detection of fat in the spleen of human subjects [10,39]. A potential explanation is that phase errors caused by eddy currents significantly alter the first echo and introduce bias into the complex-based quantification [10].

We retrospectively collected a group of twenty patients with a balanced representation of the severity grades of hepatic steatosis. The cause of liver disease was MASLD in most patients (85%). Many of the patients were obese (mean BMI: 27.5 kg/m^2^) and hardly fit into the scanner’s gantry, which could amplify the inhomogeneity of the magnetic field. To prevent artifacts from the breathing and involuntary movement of the patients, we decided to use a 2D SPGR sequence set acquisition parameters to keep acquisition time short, typically less than 30 s. The correlation was excellent across the algorithms and nearly perfect between the complex-based methods (0.934–0.999). The overall inter-rater agreement was also very good (0.891). The ANOVA showed that PDFF measured with MAG was significantly lower than MAG-R. To a small extent, this can be attributed to the spectral fat model used by MAG. However, the predominant factor is the widespread presence of phase unwrapping errors with MAG-R.

Field inhomogeneity artifacts were common and could be observed on 70% of the MAG-R and 39% of the GC images. The phase unwrapping errors produced by MAG-R typically had a speckled pattern. They were more frequent in high-grade steatosis or focal depositions, indicating that likelihood-based optimization is prone to errors when the PDFF is close to the fat/water equilibrium. Another significant drawback of MAG-R is the excessively long computation time, which can be attributed to pixel-by-pixel curve fitting and multiple initializations of the reconstruction. 

The phase unwrapping errors caused by GC mainly affected areas at the perimeter of the images or slices at the end of the stack and were more frequent in obese patients examined with a large FOV. These phase unwrapping errors result from significant inhomogeneity in the main magnetic field. A proposed solution for this problem is to apply a magnitude-based reconstruction to remove phase errors from an initial complex-based fat/water decomposition [40,41]. 

The phase unwrapping errors were absent in the QPBO reconstructions. Meanwhile, the QPBO method proved highly accurate in phantoms, and its estimates showed a high consistency with GC. The QBPO algorithm combines direct phase estimation and constraints on physical parameters with a multi-scale calculation of the parameter maps while using efficient mathematical formulation [19]. The computation time with the QPBO was also lower than with other methods except the MAG. 

Our study has several limitations: first, this was a single-center study, and patient scans were collected with one scanner; therefore, the effect of scanner type on liver PDFF estimates could not be comprehensively evaluated. Second, the components of fat/water emulsions in the two phantoms had different off-resonance spectra compared to the spectral fat model for human subjects; thus, the results in phantoms only provide a good approximation of the accuracy expected in vivo. Third, the algorithms run in MATLAB and Phyton programming environments, and multiple data processing steps must be completed to analyze MRI datasets. Fourth, we only tested the decomposition methods on scans obtained at 1.5 T field strength. The artifacts, due to field inhomogeneity and eddy currents, may be more pronounced at 3 T and higher field strengths, which may lower the accuracy of some of the methods. Fifth, the off-resonance spectrum of lipids used for the phantom experiments slightly differed from the spectrum of lipids in the human liver. We used modified spectra based on prior spectroscopy results to compensate for the bias in calculating PDFF estimates in fat–water phantoms. Sixth, the methods tested in this study are not certified for clinical use and alone cannot be used for diagnostic purposes. Seventh, liver biopsy was unavailable from all human subjects. Thus, it was not used as a reference.

## 5. Conclusions

The results of our study demonstrate that open-source software tools available in public repositories are feasible alternatives to vendor-supplied methods for fat/water decomposition and accurate estimation of PDFF on CSE-MRI datasets. The algorithms using different mathematical models provide flexibility in parameter settings and can be tailored to the user’s specific needs without changing the scanning protocol. In our hands, the MAG-R method provided the closest approximation of ground truth in an experimental setting. Meanwhile, for measuring liver PDFF in human subjects, the QPBO method was superior to its competitors in reliability and computational efficiency.

## Figures and Tables

**Figure 1 diagnostics-14-01138-f001:**
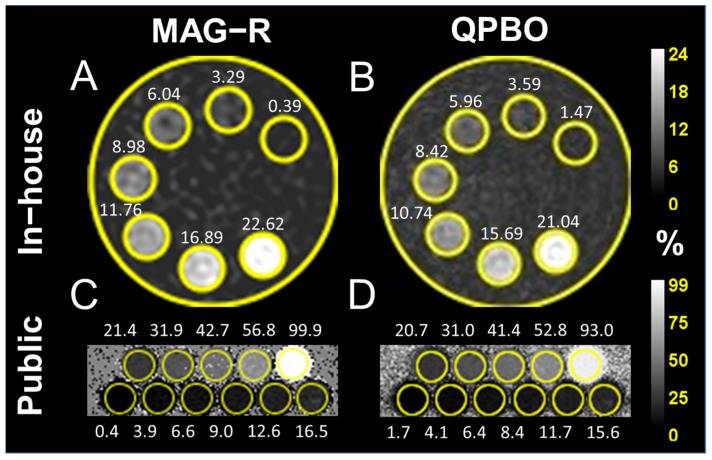
Design of the fat/water phantoms. Fat-fraction (in %) maps of the in-house built phantom were reconstructed with different algorithms, including (**A**) magnitude-only estimation with Rician noise modeling (MAG-R) and (**B**) multi-scale quadratic pseudo-Boolean optimization with graph cut (QPBO). The lower row shows the publicly available phantom dataset reconstructions with the (**C**) MAG-R and (**D**) QPBO algorithms. The scales on the left show the distribution of proton density fat fraction (PDFF) in the two phantoms.

**Figure 2 diagnostics-14-01138-f002:**
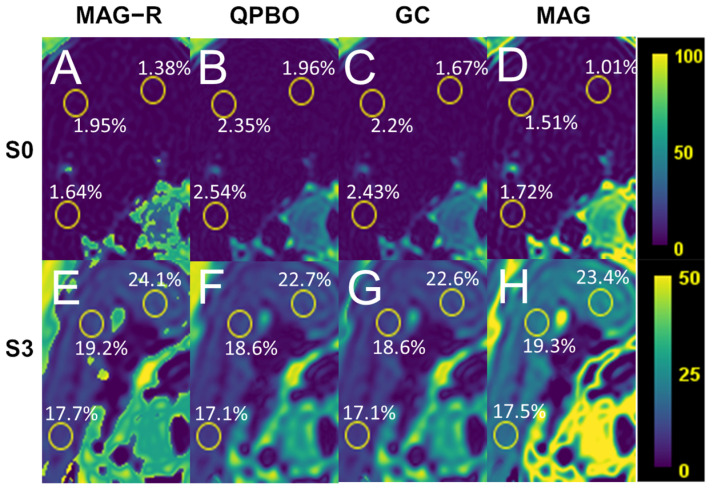
Comparison of fat fraction estimates in human livers. Four algorithms were used to reconstruct proton density fat fraction (PDFF) maps of the livers in the patients without steatosis (S0 grade) or with severe steatosis (S3 grade). The estimated PDFF was measured in three identical regions of interest (yellow circles) using (**A**,**E**) magnitude-only estimation with Rician noise modeling (MAG-R), (**B**,**F**) multi-scale quadratic pseudo-Boolean optimization with graph cut (QPBO), (**C**,**G**) graph cut (GC), and (**D**,**H**) magnitude-based (MAG) methods. The color bars on the left show the scale in unambiguous (0−100%, upper row) and MAG (0−50%, lower row) reconstructions.

**Figure 3 diagnostics-14-01138-f003:**
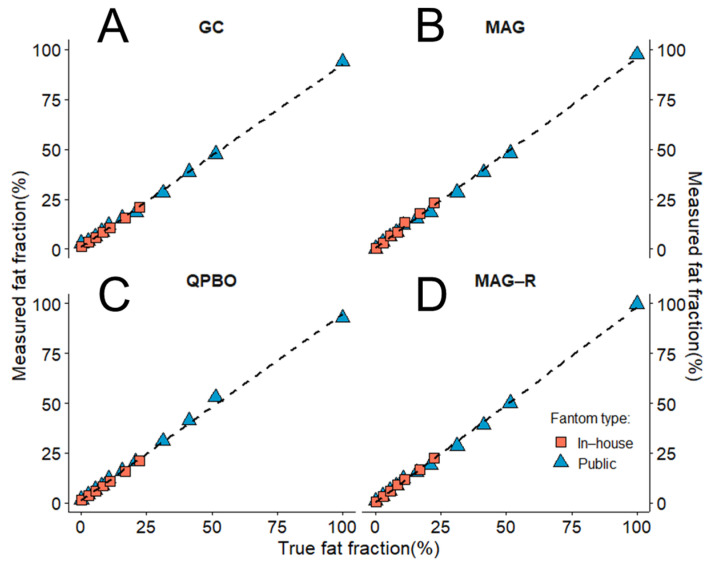
Correlation between estimated and true fat fraction in phantoms. The proton density fat fraction (PDFF) was measured in an in-house assembled fat/water phantom (rose rectangle) and a publicly available phantom dataset (blue triangle). Based on the slopes and intercepts of the regression lines (dashed lines), the estimates of the (**A**) graph cut (GC), (**B**) magnitude-based (MAG), (**C**) multi-scale quadratic pseudo-Boolean optimization with graph cut (QPBO), and (**D**) magnitude-only estimation with Rician noise modeling (MAG-R) methods showed a nearly perfect linear association with the true fat fractions.

**Figure 4 diagnostics-14-01138-f004:**
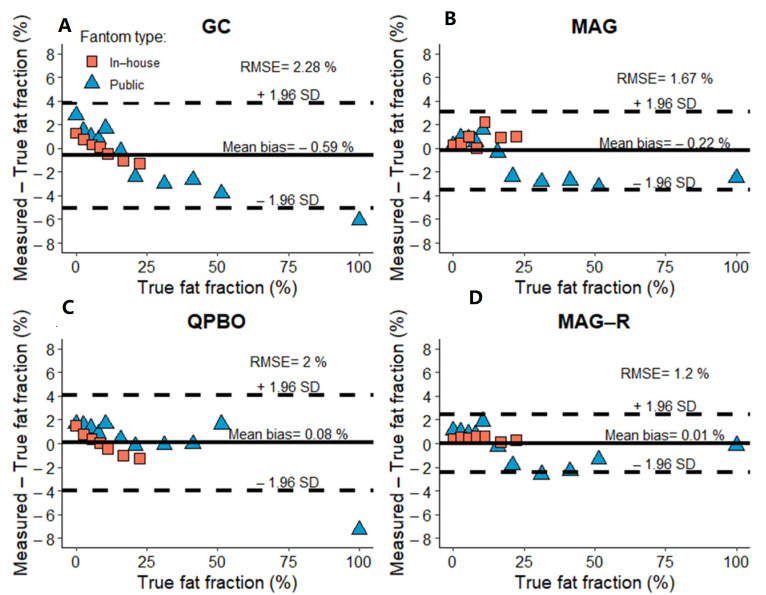
The bias of fat fraction estimates in phantoms. The Bland-Altman plots show the mean (continuous line) and the limits of the 95% confidence interval (dashed lines) of bias between estimated and true fat fractions in in-house (rose rectangle) and public (blue triangle) phantom datasets using (**A**) graph cut (GC), (**B**), magnitude-based (MAG), (**C**) multi-scale quadratic pseudo-Boolean optimization with graph cut (QPBO), and (**D**) magnitude-only estimation with Rician noise modeling (MAG-R) methods. The root mean square error (RMSE) is also displayed.

**Figure 5 diagnostics-14-01138-f005:**
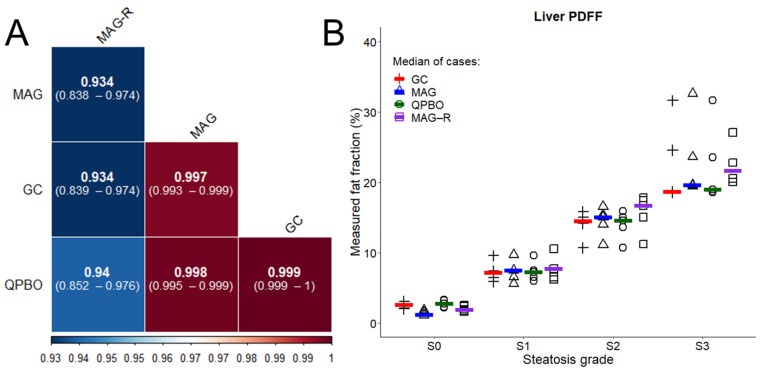
Summary of measurements in the patient group. (**A**) The correlation plot shows the Pearson r coefficients and the 95% confidence intervals obtained by pairwise comparisons across the graph cut (GC), magnitude-based (MAG), multi-scale quadratic pseudo-Boolean optimization with graph cut (QPBO), and magnitude-only estimation with Rician noise modeling (MAG-R) methods. The color bar shows the scale of the Pearson r. (**B**) The jitter plot shows the distribution of proton density fat fraction (PDFF) estimates across the steatosis severity grades (S0–S3) measured in five representative patients in each grade. The color lines display the median of the estimates in each grade using the GC (red), MAG (blue), QPBO (green), and MAG-R (purple) methods.

**Figure 6 diagnostics-14-01138-f006:**
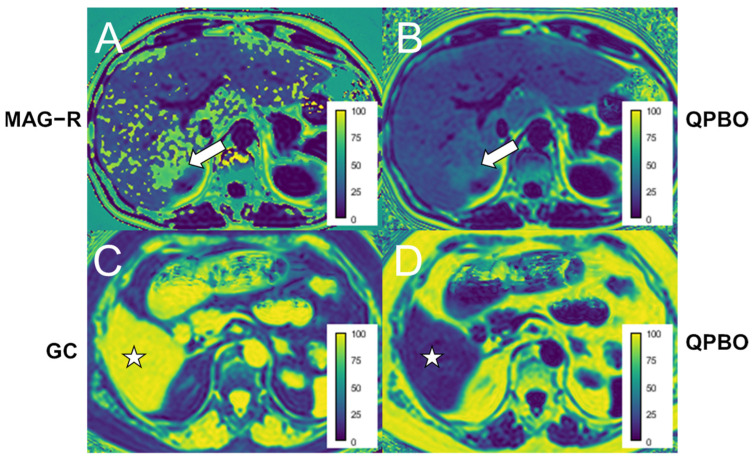
Examples of phase unwrapping artifacts on patients’ scans. (**A**) A speckled pattern of phase unwrapping errors was observed in the liver using magnitude-only estimation with Rician noise modeling (MAG-R), which was most prominent in areas with higher proton density fat fraction (PDFF), such as a focal deposition (white arrow). (**C**) The image from the end of the image stack shows that the phase is inverted in the liver (white star) using the graph cut (GC) method. (**B**,**D**) The panels show the corresponding non-erroneous PDFF estimates using multi-scale quadratic pseudo-Boolean optimization with graph cut (QPBO). The color bars display the PDFF scale in percentages.

**Table 1 diagnostics-14-01138-t001:** Demographic and clinical characteristics of the patient cohort.

Patients (*n* = 20)	*n* (%) ^1^	Mean ± SD
Age (years)	–	53 ± 15
Female/Male	9/11 (45%/55%)	–
BMI (kg/m^2^)	–	27.7 ± 4.7
T2DM	3 (15%)	–
**Etiology of liver disease ^2^**		
MASLD	15 (75%)	–
MASH	2 (10%)	–
Cardiogenic	1 (5%)	–
Hemochromatosis	1 (5%)	–
PSC	1 (5%)	–

^1^ Categorical variables are reported as number (n) and percentage, continuous variables as mean ± standard deviation. ^2^ Based on electronic medical records. BMI: body mass index, MASLD: metabolic-associated steatotic liver disease, MASH: metabolic-associated steatohepatitis, PSC: primary sclerosing cholangitis, and SD: standard deviation.

**Table 2 diagnostics-14-01138-t002:** The results of correlation, regression, and inter-rater agreement analysis with phantom measurements.

Method	Correlation ^1^	*p*-Value		
GC	0.999 (0.997–1)	<0.001		
QPBO	0.999 (0.996–0.999)	<0.001		
MAG	0.999 (0.996–1)	<0.001		
MAG-R	0.999 (0.997–1)	<0.001		
**Method**	**Slope** **^2^**	**Intercept** **^2^**	**R ^2^**	***p*-Value**
GC	1.09	−1.13	0.998	<0.001
QPBO	1.07	−1.43	0.997	<0.001
MAG	1.05	−0.71	0.997	<0.001
MAG-R	1.02	−0.45	0.998	<0.001
**Method**	**ICC** **^3^**	***p*-Value**		
GC	0.995 (0.988–0.998)	<0.001		
QPBO	0.996 (0.991–0.999)	<0.001		
MAG	0.998 (0.994–0.999)	<0.001		
MAG-R	0.999 (0.997–1)	<0.001		

^1^ Pearson r coefficient with a true fat fraction with 95% confidence intervals. ^2^ Values from the linear regression analysis with true fat fraction as the dependent and measured fat fraction as a predictor variable. ^3^ Intra-class correlation coefficient reported with 95% confidence interval. GC: graph cut, QPBO: multi-scale quadratic pseudo-Boolean optimization with graph cut, MAG: magnitude-based, and MAG-R: magnitude-only estimation with Rician noise modeling.

## Data Availability

Measurement data are included in the Appendix A.

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
