# Peer review of "Comparison of Vendor-Independent Software Tools for Liver Proton Density Fat Fraction Estimation at 1.5 T"

_diagnostics, 2024, doi:10.3390/diagnostics14111138_

Round 1
Reviewer 1 Report
Comments and Suggestions for Authors
There are some comments.
-
To verify the accuracy of the tests, comparative studies with biopsy tissue is recommended.
-
It would be better to describe the content of the photos in Figures 1 and 2 in a simple and detailed way.
-
It would be better to clarify the criteria for the degrees of steatosis (S0-S3).
Author Response
To verify the accuracy of the tests, comparative studies with biopsy tissue is recommended.
We agree with the reviewer that liver biopsy is the gold standard diagnostic modality of liver steatosis. However, liver biopsy is not regularly performed in MASLD, and it is only required if the etiology of liver disease is unknown or the clinical findings indicate concomitant steatohepatitis. In our institution, we do not use liver biopsy to detect hepatic steatosis in uncomplicated cases of MASLD due to its higher patient burden and risk of complications. Also, liver biopsy only samples a small portion of the liver parenchyma; therefore, its reproducibility is low for mild and moderate steatosis. Consequently, we prefer non-invasive tests to diagnose significant hepatic steatosis, and tissue samples from all human subjects were unavailable. Also, the primary goal of the present study was not to determine the diagnostic accuracy of MRI-PDFF but rather to reveal a systemic bias between the different fat/water decomposition methods in human subjects. We added this explanation to the study's limitations (please see rows 512-513 on page 13).
It would be better to describe the content of the photos in Figures 1 and 2 in a simple and detailed way.
We are uncertain regarding the reviewer's recommendation. We shortened the descriptions of Figure 1 and Figure 2.
It would be better to clarify the criteria for the degrees of steatosis (S0-S3).
We have added the definitions of steatosis grades to the Methods section (please see rows 208-209 on page 5).
Reviewer 2 Report
Comments and Suggestions for Authors
This report compares 4 open-source software tools to estimate PDFF using chemical-shift encoded MRI. The tools were tested on phantoms with known fat/water ratios and patients with varying degrees of fatty liver. All algorithms accurately estimated PDFF in phantoms, but there were differences in their performances. The magnitude-only estimation with MAG-R method was the most accurate, while the multi-scale quadratic pseudo-Boolean optimization with graph cut (QPBO) method was the most reliable and efficient in patients. Overall, this study demonstrates that open-source software tools can be promising alternatives to vendor-supplied methods for PDFF estimation.
However, there are several minor concerns to be addressed concisely:
- The manuscript lacks citations for several claims and statements, including the prevalence of MASLD, the relationship between fatty liver disease and cardiovascular morbidity, and the diagnostic criteria for hepatic steatosis.
- Please specify the type of ANOVA (one-way, two-way, etc.) or the post-hoc tests used for pairwise comparisons. Additionally, the manuscript should clarify whether the p-values reported in the results section are raw or adjusted for multiple comparisons.
- The limitations section could discuss the potential impact of using different field strengths (e.g., 3T) or the limitations of using phantoms with different fat spectra than human livers.
- The manuscript uses the term "fat/water swap" to describe mapping errors. "Phase unwrapping errors" or "field inhomogeneity artifacts" would be more appropriate.
Comments on the Quality of English LanguageLanguage and Style: There are several minor grammatical errors throughout the current report.
Author Response
This report compares 4 open-source software tools to estimate PDFF using chemical-shift encoded MRI. The tools were tested on phantoms with known fat/water ratios and patients with varying degrees of fatty liver. All algorithms accurately estimated PDFF in phantoms, but there were differences in their performances. The magnitude-only estimation with MAG-R method was the most accurate, while the multi-scale quadratic pseudo-Boolean optimization with graph cut (QPBO) method was the most reliable and efficient in patients. Overall, this study demonstrates that open-source software tools can be promising alternatives to vendor-supplied methods for PDFF estimation.
We thank reviewer 2 for the favorable comments on our manuscript!
However, there are several minor concerns to be addressed concisely:
Question 1: The manuscript lacks citations for several claims and statements, including the prevalence of MASLD, the relationship between fatty liver disease and cardiovascular morbidity, and the diagnostic criteria for hepatic steatosis.
In the revised manuscript, we have addressed the reviewer’s concerns and included a more recent definition of fatty liver disease from the AASLD guideline (Please see rows 64-66 on page 2). We also added more recent references on the prevalence of MASLD in developed countries (Please see rows 35-37 on page 1) and the higher incidence of CVD associated with MASLD and MetALD during long-term follow-up (Please see rows 39-41 on page 1).
Question 2: Please specify the type of ANOVA (one-way, two-way, etc.) or the post-hoc tests used for pairwise comparisons. Additionally, the manuscript should clarify whether the p-values reported in the results section are raw or adjusted for multiple comparisons.
To compare liver PDFF estimates across the methods, we used a two-way paired ANOVA test and a two-way paired-sample t-test for post hoc analysis. The Bonferroni correction adjusted p-values for multiple pairwise post hoc comparisons. This information is now included in the Methods section (please see rows 300-303 on page 8 ).
Question 3: The limitations section could discuss the potential impact of using different field strengths (e.g., 3T) or the limitations of using phantoms with different fat spectra than human livers.
We agree with the reviewer's comment that field inhomogeneity can lower the accuracy of the PDFF estimates at higher field strengths. We added to the study's limitations the lack of comparison between 1.5T and 3T acquisitions. (please see rows 505-508 on page 13). The off-resonance spectrum of lipids used in the phantoms slightly differed from lipids in the human liver. We used a modified spectrum based on previously reported spectroscopy data to compensate for the bias in calculating PDFF estimates in phantoms. This has also been added to the study's limitations (please see rows 508-511 on page 13).
Question 4: The manuscript uses the term "fat/water swap" to describe mapping errors. "Phase unwrapping errors" or "field inhomogeneity artifacts" would be more appropriate.
As suggested by the reviewer, we have revised the terminology and replaced fat/water swap with phase unwrapping errors in the manuscript.